# Experimental Investigation of Tip Wear of AFM Monocrystalline Silicon Probes

**DOI:** 10.3390/s23084084

**Published:** 2023-04-18

**Authors:** Song Huang, Yanling Tian, Tao Wang

**Affiliations:** 1School of Mechanical Engineering, Tianjin University, Tianjin 300350, China; 2020201199@tju.edu.cn (S.H.); 2021201180@tju.edu.cn (T.W.); 2School of Engineering, University of Warwick, Coventry CV4 7AL, UK

**Keywords:** AFM probe, probe wear, nanomachining, wear process, machining quality

## Abstract

AFM has a wide range of applications in nanostructure scanning imaging and fabrication. The wear of AFM probes has a significant impact on the accuracy of nanostructure measurement and fabrication, which is particularly significant in the process of nanomachining. Therefore, this paper focuses on the study of the wear state of monocrystalline silicon probes during nanomachination, in order to achieve rapid detection and accurate control of the probe wear state. In this paper, the wear tip radius, the wear volume, and the probe wear rate are used as the evaluation indexes of the probe wear state. The tip radius of the worn probe is detected by the nanoindentation Hertz model characterization method. The influence of single machining parameters, such as scratching distance, normal load, scratching speed, and initial tip radius, on probe wear is explored using the single factor experiment method, and the probe wear process is clearly divided according to the probe wear degree and the machining quality of the groove. Through response surface analysis, the comprehensive effect of various machining parameters on probe wear is determined, and the theoretical models of the probe wear state are established.

## 1. Introduction

With the continuous development of science and technology, nanotechnology has become more and more mature. The appearance of the atomic force microscope has greatly promoted the development of nanotechnology and has become the main scientific research tool to explore this field. At present, AFM has been widely used in surface science, materials science, electrochemistry, biology, metrology, and other fields. By controlling the nanodistance between the probe and the sample and adjusting the atomic force between them, AFM can be used to complete the scanning imaging of the nanostructure [1], and realize the machining of grooves, nanogratings and nanoflow controlled channels [2,3,4], functional surfaces, and complex nanopatterns [5,6,7,8], and the manufacturing of micro-nano electronic components [9,10]. Currently, the common morphological characterization methods of AFM probes include SEM and TEM observation [11,12,13], reconstruction and blind reconstruction [14,15,16,17], adhesion characterization [11,13], and the nanoindentation method [18,19,20]. In AFM nanomachining, tapping monocrystalline silicon probes are often used because the cantilever stiffness of the probes is larger and they can bear larger normal loads. The wear of the monocrystalline silicon probe becomes particularly significant under the driving force. At present, many researchers have carried out detailed studies of the wear mechanism, the wear process division of monocrystalline silicon probes, and the wear effects of AFM machining parameters.

The probe wear mechanism includes adhesive wear, fatigue wear, chemical wear, fracture wear, oxidation wear, and friction wear. It is related to the material properties of the probe and the sample, the experimental environment, and the stress state between the probe and the sample; furthermore, there are differences in the probe wear mechanism at micro and macro scales. Jacobs [21] believed that the wear mechanism of monocrystalline silicon probe scanning on diamond samples under vacuum conditions was chemical wear. Chung [22] conducted probe wear experiments with monocrystal silicon probes in atmospheric and pure nitrogen environments, and found that the wear mechanism of monocrystal silicon probes in atmosphere was oxidative wear. Gotmann et al. [23] believed that the wear mechanism of monocrystalline silicon probe during scanning on polymer samples was friction wear, and silicon atoms would separate from the matrix atom by atom.

The probe wear process is divided according to the wear degree, the wear speed, and the effect of the probe wear on the scanning and machining of the nanostructure. Kong [11] explored the probe wear condition in a vibration-assisted machining system of AFM. They divided the probe wear process into the initial wear stage, transition stage, and failure stage, and reasonably predicted the service life of the probe.

At the same time, the scanning and machining parameters of AFM also have a significant influence on probe wear. Chung [24] studied the wear condition of monocrystal silicon probes with different tip radii under different normal forces, sliding speeds, and sliding distances. They found that the wear rate increased with the increase in normal force and sliding velocity, and decreased with the increase in initial tip radius. Flater [25] used a monocrystalline silicon probe to conduct a sliding experiment on an alumina sample and found that the contact stress between the probe and the sample had an exponential relationship with the wear rate. Mukhtar [26] studied the wear of the monocrystalline silicon probe in the process of scratching on a monocrystalline copper sample with different normal loads, sliding distances, and sliding directions. They qualitatively analyzed the evolution of the AFM tip profile at different sliding distances and sliding directions, and quantified the wear tip radius and wear volume. They found that the probe wear volume increased with the increase in normal load, and the tip was prone to fracture wear under a larger load. Xu [27] studied the influence of scanning parameters on AFM probe wear in scanning mode. They experimentally investigated the effects of normal force and scanning speed on tip wear, and found that most of the volume loss was caused by tip fracture. In the scanning process, the transverse dynamic load on the probe can easily lead to tip fracture.

Morphological characterization methods applied to AFM probes are often used for regular tip morphology detection, which will mostly lead to tip wear in the detection process. The AFM probe tips become irregular after wear, and most of them have an approximately spherical structure. Most of the studies of the wear of monocrystalline silicon probes focus on probe wear in the process of nanoscale measurement. Obviously, probe wear is more significant in the process of nanomachining, and the effect of probe wear on the nanomachining quality is more intuitive. In this paper, the wear condition of the monocrystalline silicon probe in the process of nanomachining is studied. The wear tip radius is detected by the nanoindentation Hertz model characterization method, and the detection results are verified by SEM observation. The wear tip radius, wear volume, and probe wear rate are used as the evaluation indexes of the probe wear state. The effects of scratching distance, normal load, scratching speed, and initial tip radius on the probe wear are analyzed by a single factor experiment, and the probe wear process is divided according to the probe wear degree and the machining quality of the groove. Through response surface analysis, the comprehensive effect of various machining parameters on probe wear is determined, and theoretical models of the probe wear state are established.

## 2. Materials and Methods

In this study, nanomanipulation was performed under the CSPM5500 model AFM produced by China Being Company. The Tap300-G monocrystalline silicon probe was used for groove scratching on the hard PVC substrate, and nanoindentation measurement on the soft PVC substrate. The wear tip radius was detected by nanoindentation Hertz model characterization method, and the wear volume and probe wear rate were calculated based on the wear tip radius. Figure 1 shows the morphology of a Tap300-G monocrystalline silicon probe without wear. It can be seen from the figure that the pointy part of the probe is conical, with a half cone angle of 10°, while the other part is pyramidal, with a half cone angle of about 25°. The probe tip radius is about 20 nm. The substrate materials used for nanoscratching machining and nanoindentation measurement are hard PVC and soft PVC, respectively. These two kinds of polymer material have good mechanical properties, which are suitable for nanomanipulation, and are common and easy to obtain.

The Hertz model is often used in the case of elastic deformation under spherical structure contact, and does not consider the effect of adhesion between the probe and the substrate. In this paper, the nanoindentation operation is carried out on the polymer substrate with the monocrystalline silicon probe, the indentation loading curve is determined, and the data of the elastic deformation section in the indentation loading curve is fitted by the Hertz model. Most of the wear probes obtained by nanoscratching on the polymer substrate have spherical tips. Moreover, the elastic modulus of the polymer material is small and the elastic deformation interval is large, so it is easy to collect the fitting data. The AFM probe used is a tapping probe with a high cantilever stiffness of 40 N/m, so the effect of adhesion on the nanoindentation data can be ignored. Figure 2 shows the nanoindentation diagram of the spherical tip. Under the normal load, the tip is pressed vertically into the substrate and produces the elastic compression depth on the substrate. The spherical tip radius can be calculated based on the Hertz model.

In the Hertz model, the normal load on the spherical tip and the elastic pressing depth satisfy Equation (1).
(1)$$F=\frac{4}{3}E^{*}R^{1/2}\delta^{3/2}$$
where *F* is the normal load on the probe tip. *E** is the composite modulus of the probe and substrate; *R* is the probe tip radius; *δ* is the elastic pressing depth. *E** is determined by the mechanical parameters of both the probe and the substrate material, and given by Equation (2).
(2)$$E^{*}=\frac{E_{s}}{\left(1-v_{s}^{2}\right)+\left(1-v_{t}^{2}\right)E_{s}/E_{t}}$$
where *E_s_* and *v_s_* are the elastic modulus and Poisson’s ratio of the substrate material. respectively, and *E_t_* and *v_t_* are the elastic modulus and Poisson’s ratio of the probe material, respectively. If the elastic modulus of the probe material is much larger than that of the substrate material, then $\left(1-v_{t}^{2}\right)E_{s}/E_{t}\to 0$ and thus the composite modulus can be obtained by:(3)$$E^{*}\approx \frac{E_{s}}{1-v_{s}^{2}}$$

The fitting coefficient *α* can be obtained by using the Hertz model to fit the data of the elastic deformation section in the nanoindentation loading curve. The fitting coefficient *α* is defined by:(4)$$\alpha =\frac{4}{3}E^{*}R^{1/2}$$

The fitting coefficient *α* is determined by the composite modulus *E** and the tip radius *R*. In the case that the mechanical properties of the probe and the substrate material are known, the tip radius *R* can be estimated by:(5)$$R={\left(\frac{3\alpha }{4E^{*}}\right)}^{2}$$

It is well known that the probe wear rate *k* is affected by the normal load and the scratching distance. The probe wear rate *k* can be calculated by:(6)$$k=\frac{\Delta V}{FL}$$
where ∆*V* is the probe wear volume, which can be estimated by changes in the probe tip radius, *F* is the normal load on the probe, and *L* is the scratching distance.

## 3. Results

### 3.1. Experimental Study on Wear Effect of AFM Machining Parameters

Ten monocrystalline silicon probes of the Tap300-G model were selected to conduct groove scratching on a hard PVC substrate. Figure 3 shows the grooves machined using the unworn AFM probe. The length of the groove is 20 μm, and the interval of each groove is 2 μm. The normal load applied to the probe varies in the range of 0~6.4 μN, and the scratching speed can be adjusted by changing the scratching duration of the grooves. The experiment was repeated three times under the same experimental conditions according to the machining parameters in Table 1. The wear tip radius detected in three wear experiments was averaged, and the wear volume and probe wear rate were calculated using the mean value of the wear tip radius.

According to the single factor experimental design, the nanomachining parameters of 10 probes were set as shown in Table 1. The tip radius of the wear probe was detected by the nanoindentation Hertz model characterization method. Probes 1~5 were used to study the effect of the normal load on probe wear. The scratching speed and the scratching distance were controlled uniformly, and the normal load was set to increase linearly in the range of 1.6~6.4 μN. Probes 6~10 were used to study the effect of scratching speed on probe wear. The normal load and scratching distance were controlled uniformly, and the scratching speed was set to increase linearly in the range of 2~20 μm/s.

The nanoindentation Hertz model characterization method can realize the in situ detection of the wear probe tip radius, so the wear tip radius can be detected quickly and conveniently during the wear process of the probe. In the experiment, the wear tip radius was detected using the nanoindentation Hertz model characterization method for every interval of a certain sliding distance, and the wear volume and probe wear rate were calculated so as to accurately grasp the wear state of the probe.

#### 3.1.1. Probe Wear State Detection

After the probes completed the total scratching stroke of 40 mm, the wear probes were placed in SEM to observe the wear tip morphology. Figure 4 shows the tip morphologies of the ten wear probes and the enlarged view of the tip of the worn probe is shown in the upper right section. As can be seen from the figure, the tip profile of these wear probes is approximately spherical, and the pointy part with a cone angle of 10° has been worn away. A large amount of debris adheres to the sidewalls of Probe 4 and Probe 5, indicating that a lot of material accumulated on the substrate adheres to the probe surface under the larger normal load. Other probes have smooth sidewalls with little detrital adhesion. The spherical characteristic is used to fit the worn tip profile, which has a good fitting effect. The fitted spherical tip radius can be measured. It can be seen from the figure that with the increase in the normal load or scratching speed, the measured value of the wear tip radius gradually increases.

These wear probes were used to perform the nanoindentation operation on soft PVC sheets. The force response curve was collected using the curve measurement function on the AFM, and the indentation loading curve was obtained by data processing. Before the indentation test, the material properties of the substrate were measured by the nanoindentation instrument, and the composite modulus of the probe–substrate was measured to be 0.053 Gpa. The elastic deformation range of the substrate was determined to be 0~20 nm by finite element simulation of nanoindentation. Figure 5 shows the Hertz model fitting of indentation data for wear Probe 2. The Hertz model can be used to fit the data of the elastic deformation section of the indentation loading curve to obtain the fitting coefficient, and the wear tip radius can be calculated based on Equation (5). Table 2 shows the calculation results and error analysis of the wear tip radius of Probes 1~10. The wear tip radius was calculated using the nanoindentation Hertz model characterization method. Compared with the tip radius measurement using the SEM observation method, the error values were all within 15%. In conclusion, it is feasible to calculate the wear tip radius using the nanoindentation Hertz model characterization method.

#### 3.1.2. Scratching Distance Effect

In the experiment, the probe wear state was detected with a certain scratching distance at every interval, and the wear tip radius, the wear volume, and the probe wear rate were calculated. The wear tip radius detected in repeated experiments was statistically processed to calculate its mean value and standard deviation. Figure 6 shows the relationship between the scratching distance and the wear probe tip radius. As can be seen from the figure, the wear probe tip radius gradually increases with the increase in the scratching distance, and the wear probe tip radius changes the most within the range of 0~5 mm.

Figure 7 shows the relationship between the scratching distance and the probe wear volume. It can be seen that the probe wear volume gradually increases with the increase in the scratching distance, and the growth rate of the probe wear volume is relatively large within the range of 0~20 mm. Due to the high normal load and scratching speed set on Probe 5 and Probe 10, both tip radius and wear volume increased sharply in the range of 0~5 mm, that is, the probe appears to experience fracture wear.

Figure 8 shows the relationship between the scratching distance and the probe wear rate. It can be seen that with the increase in the scratching distance, the probe wear rate gradually decreases and eventually becomes stable. In the range of 0~20 mm, the probe wear rate decreased rapidly. With the increase in the scratching distance, the wear probe tip radius increases gradually, and the contact stress between the probe and the substrate decreases gradually, so that the probe wear rate decreases gradually. It can also be seen that the smaller the normal load, the greater the probe wear rate, and the scratching speed has little effect on the probe wear rate.

#### 3.1.3. Normal Load Effect

Figure 9 shows the relationship between normal load and tip radius, wear volume, and wear rate. As can be seen from the figure, with the increase in normal load, the wear probe tip radius increases gradually, but its growth rate decreases first and then increases sharply. The variation in probe wear volume is similar. In the range of 0~4.8 μN, the probe wear mechanism is friction wear and the probe wear rate decreases gradually. In the range of 4.8~6.4 μN, the probe appears to experience fracture wear, so that the wear rate increases.

#### 3.1.4. Scratching Speed Effect

Figure 10 shows the relationship between the scratching speed and tip radius, wear volume, and wear rate. As can be seen from the figure, the wear probe tip radius and the probe wear volume increased slowly with the increase in the scratching speed. In the range of 0~14 μm/s, the probe wear mechanism is friction wear and wear rate remains stable. In the range of 14~20 μm/s, the probe wear rate increases rapidly, and the probe appears to experience fracture wear.

#### 3.1.5. Effect of Initial Tip Radius

The wear tip radius of Probe 2 was recorded every sliding distance interval of 10 mm, and was used as the initial tip radius of the probe. Furthermore, the wear volume and probe wear rate during the subsequent 10 mm sliding process were calculated, and the influence of the initial tip radius on the probe wear was analyzed. Figure 11 shows the relationship between the initial tip radius and the probe wear volume and probe wear rate. As can be seen, with the increase in the initial tip radius, the probe wear volume and the probe wear rate decreased linearly. The main reason for this is that the increase in the probe tip radius greatly reduces the contact stress between the probe and the substrate, which slows down the wear of the probe, and then leads to the change in the wear process from the initial wear stage to the stable wear stage.

#### 3.1.6. Investigation of Probe Wear Process

The new AFM probes were used to scratch hard PVC sheets with limited machining parameters. The normal load was set to 6.4 μN and the scratching speed was set to 20 μm/s. The effects of the scratching distance on the wear tip radius and the scratching depth of the grooves were analyzed to study the wear process of the probe.

Figure 12 shows the morphology and cross-section data of grooves scratched by six wear probes with different tip radii. As can be seen from the figure, with the increase in the wear tip radius, the depth and width of the machined grooves and the material accumulation on the sample surface gradually decreased. That is, the machining capability of the AFM probe gradually weakened. At the same time, the machining quality gradually deteriorated. When the wear tip radius is less than 100 nm, the machining quality of the grooves is better.

Figure 12c shows how the wear probe tip radius and the depth of the machined grooves change with the scratching distance. It can be seen that with the increase in scratching distance, the wear probe tip radius gradually increases, and the scratching depth of the groove gradually decreases and eventually tends to zero. In the range of 0~200 mm, the wear probe tip radius and the scratching depth of the grooves change relatively significantly. When the scratching distance reaches 2000 mm, the wear probe tip radius is about 400 nm, and the scratching depth of the grooves is about 10 nm, and the nanomachining ability of the probe is almost lost.

### 3.2. Theoretical Modeling of Wear Effects of AFM Machining Parameters

In order to quantify the influence of nanomachine parameters on the probe wear and establish the theoretical model of the probe wear state, the response surface experiment was designed and carried out. In the experiment, three quantitative factors were set, namely, normal load, scratching speed, and scratching distance. In order to obtain better groove machining quality and reduce the influence of probe wear on machining quality, appropriate machining parameters should be selected. Through the qualitative analysis of the machining parameters, it can be concluded that when the scratching distance is 20~40 mm, the normal load is 1.6~6.4 μN, and the scratching speed is 2~20 μm/s, it is the most suitable for nanomachining. In order to prevent fracture wear of the probe, it is necessary to avoid using a large normal load and scratching speed. Therefore, the normal load ranged from 1.6 μN to 4.8 μN, the scratching speed ranged from 2 μm/s to 14 μm/s, and the scratching distance ranged from 20 mm to 40 mm in the experiment. The data type of the three factors was set to continuous data. Three responses were set, namely, the wear tip radius, the wear volume, and the probe wear rate. The response surface experiment scheme and results are shown in Table 3.

#### 3.2.1. Significance Analysis of Wear Factors

Both wear tip radius and wear volume can reflect the probe wear degree, while the probe wear rate can reflect the wear speed of the probe. The fitting model used in response surface analysis is mostly the quadratic function model, which not only considers the influence of the normal load, scratching speed, and scratching distance on the responses, but also consider the interaction between these three factors and the influence of quadratic factors. The formula form of the quadratic function model is shown as follows:(7)$$z=z_{0}+ax+by+cm+dxy+exm+fym+gx^{2}+ky^{2}+lm^{2}$$
where *z*_0_, *a*, *b*, *c*, *d*, *e*, *f*, *g*, *k*, and *l* are the coefficients of the quadratic regression model, *x* is the normal load factor, *y* is the scratching speed factor, *m* is the scratching distance factor, and *z* is the response.

Next, analysis of variance was carried out to analyze the validity of the experimental results and evaluate the importance of each factor. By analyzing the contribution of variance from different sources to the total variance, the influential significance of single or multiple factors on the experimental results is determined. Table 4 shows the results of analysis of variance. It can be seen from the table that in the three responses, the *p* values of the three main factors, namely, normal load, scratching speed, and scratching distance, are all less than 0.05. It can be seen that they all have significant effects on wear tip radius, wear volume, and probe wear rate. It can be seen from the *F* values that for the wear tip radius response, wear volume response, and probe wear rate response, normal load factor has the greatest influence, followed by the scratching distance factor, and then the scratching speed factor. At the same time, the quadratic term of normal load also has a significant effect on the wear tip radius response and wear volume response, but the effect is smaller than that of the main factor. The cross term of normal load and scratching distance and the quadratic term of normal load also have significant influence on probe wear rate response, and the influential significance is higher than that of scratching speed factor. In addition, the influence of other quadratic factors is not significant, so these factors are not introduced in the variance table.

#### 3.2.2. Response Surface Analysis and Theoretical Modeling

Through response surface analysis, the comprehensive effects of various factors on the responses can be determined, and the function models between the factors and the responses can be obtained. Figure 13, Figure 14 and Figure 15, respectively, show the three-dimensional response surfaces of wear tip radius, wear volume, and probe wear rate. The normal load factor and scratching speed factor are the *x* axis and *y* axis variables, respectively. The scratching distance is 30 mm level and the surface color reflects the size of the response value. As can be seen from the figure, the increase in normal load and scratching speed will lead to the increase in wear tip radius and wear volume, but the influence of normal load is more significant than that of scratching speed. In the range of 1.6~3.2 μN, wear tip radius and wear volume increase rapidly. The increase in normal load leads to the decrease in the probe wear rate, but the effect of scratching speed on probe wear rate is not obvious. In the range of 1.6~3.2 μN, the probe wear rate decreases rapidly with the increase in normal load. In the experiment, the wear tip radius varies from 55 nm to 80 nm, and the wear volume varies from 6.2 × 10^5^ nm^3^ to 1.1 × 10^6^ nm^3^. The probe wear rate varies from 5.0 × 10^−6^ mm^3^/(Nm) to 2.0 × 10^−5^ mm^3^/(Nm).

The quadratic regression model was used to fit the experimental data, and the theoretical formulas of wear tip radius, wear volume, and probe wear rate were obtained. The coefficients of the theoretical formula are shown in Table 5. By comparing the theoretical and actual responses, the error range of the theoretical model of wear tip radius is ±1 nm, the error range of the theoretical model of wear volume is ±1.5 × 10^4^ nm^3^, and the error range of the theoretical model of probe wear rate is ±4.2 × 10^−7^ mm^3^/(Nm). The error values of the three theoretical models are small, so the theoretical model established is accurate and reliable.

## 4. Discussion

In the study of the characterization method of the wear probe tip radius, the nanoindentation Hertz model has high accuracy, and can characterize the tip radius of wear probes with spherical or nearly spherical tips. The greater the fit degree between the tip profile of wear probes and spherical characteristics, the higher the accuracy of the calculated tip radius.

In the study of the probe wear process, the probe wear process is divided into three parts, namely, the initial wear stage, the stable wear stage, and the failure stage. The initial wear stage is when the scratching distance ranges from 0 mm to 20 mm. In this stage, the probe tip radius and probe wear volume increase rapidly, and there is a peak wear rate. When normal load or scratching speed is set too high, fracture wear will occur on the probe. When the scratching distance is 20~2000 mm, it is the stable wear stage. The probe tip radius and probe wear volume increase steadily, and the wear rate tends to be stable, approximately 5.0 × 10^−6^ mm^3^/(Nm). When the scratching distance is greater than 2000 mm, the probe almost loses the capability of nanoscratching on hard PVC sheets.

In the study of the wear effect of AFM machining parameters, the effect of machining parameters on probe wear was analyzed qualitatively and quantitatively. With the increase in the scratching distance, the wear tip radius and the wear volume gradually increase, and the probe wear rate gradually decreases and eventually tends to be stable. With the increase in normal load, the wear tip radius and wear volume increase gradually, and the probe wear rate decreases first and then becomes stable. With the increase in the scratching speed, the wear tip radius and the wear volume increase slowly, and the probe wear rate remains stable. The comprehensive effects of normal load, scratching speed, and scratching distance on the probe wear state were studied using a response surface experiment. Through analysis of variance, it was determined that three main factors have significant effects on the probe wear state. Among them, the normal load has the most significant effect, followed by the scratching distance, and finally the scratching speed. Some quadratic factors also have significant effects on probe wear. Through response surface analysis and quadratic regression model data fitting, the theoretical models of wear tip radius, wear volume, and probe wear rate were established, and the error analysis of the theoretical models was carried out.

## 5. Conclusions

In this paper, the wear tip radius was measured by the nanoindentation Hertz model characterization method. Moreover, the SEM observation method was used to verify the measurement results, and the errors were all within 15%. Therefore, the nanoindentation Hertz model characterization method can realize the rapid and accurate calibration of the wear tip radius. The characterization method is suitable for worn probes with a tip radius of more than 50 nm, and with spherical or nearly spherical tip profiles.

The probe wear process is divided into three stages according to the probe wear degree and the machining quality of the groove, namely, the scratching distance of 0~20 mm is the initial wear stage, the scratching distance of 20~2000 mm is the stable wear stage, and the scratching distance above 2000 mm is the failure stage.

The effects of scratching distance, normal load, scratching speed, and initial tip radius on probe wear were qualitatively analyzed using the single factor experiment method. With the increase in the scratching distance, the wear tip radius and wear volume increase gradually, and the probe wear rate decreases gradually and finally tends to be stable. With the increase in normal load, the wear tip radius and wear volume increase gradually, and the probe wear rate decreases first and then becomes stable. With the increase in the scratching speed, the wear tip radius and the wear volume increase slowly, and the probe wear rate remains stable. When the normal load is 4.8~6.4 μN or the scratching speed is 14~20 μm/s, the fracture wear of the probe will occur in the initial wear stage, which leads to the rapid increase in the wear tip radius, wear volume, and probe wear rate. In the stable wear stage, the wear mechanism of the probe is friction wear. The smaller the normal load is set, the greater the probe wear rate, and the scratching speed has little effect on the probe wear rate.

The comprehensive effects of normal load, scratching speed, and scratching distance on the probe wear state were quantitatively analyzed using a response surface experiment. Through analysis of variance, it was determined that three main factors have significant effects on the probe wear state. Among them, the normal load has the most significant effect, followed by the scratching distance, and finally the scratching speed. Some quadratic factors also have significant effects on probe wear. Through quadratic regression model data fitting, the theoretical models of wear tip radius, wear volume, and probe wear rate were established. When the scratching distance is 20~40 mm, the normal load is 1.6~6.4 μN, and the scratching speed is 2~20 μm/s. Based on the theoretical models, the precise control of the probe wear degree and the probe wear rate can be realized through the scratching distance, normal load, and scratching speed within the set range. Thus, it can ensure the reliability of nanomachining quality.

## Figures and Tables

**Figure 1 sensors-23-04084-f001:**
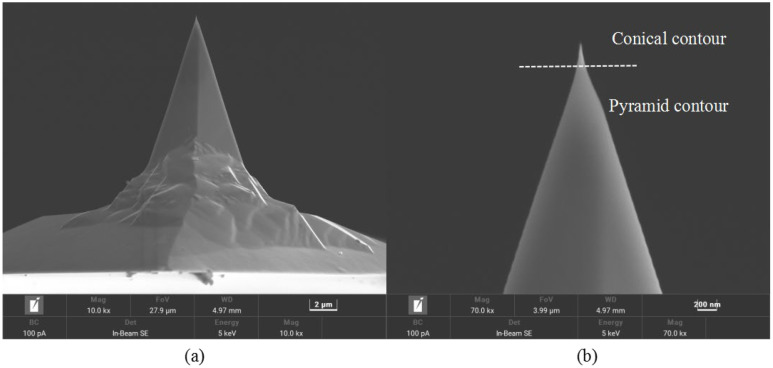
Unworn probe tip morphology: (**a**) overall image of the probe; (**b**) locally enlarged image of the tip.

**Figure 2 sensors-23-04084-f002:**
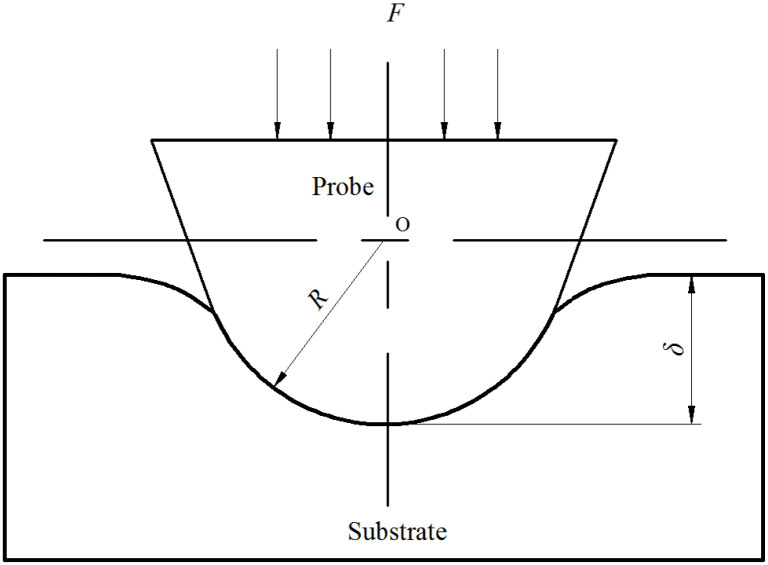
Nanoindentation diagram of the spherical tip.

**Figure 3 sensors-23-04084-f003:**
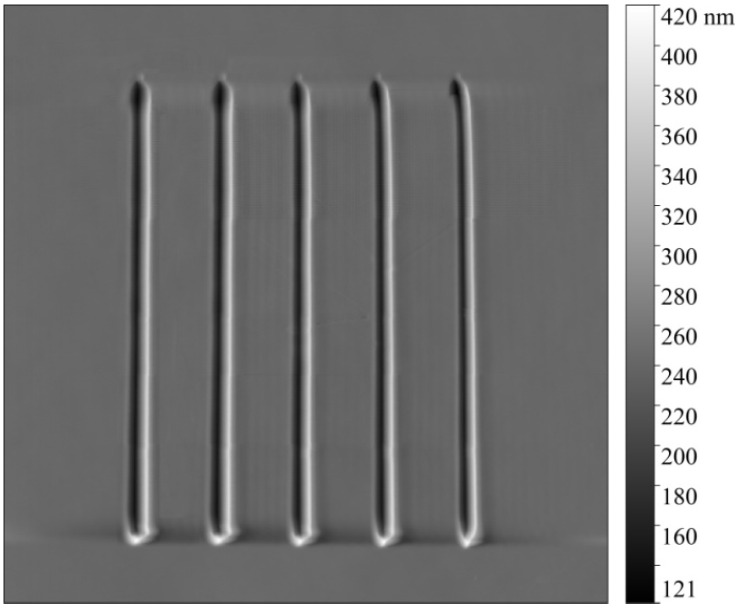
Grooves machined by an unworn AFM probe.

**Figure 4 sensors-23-04084-f004:**
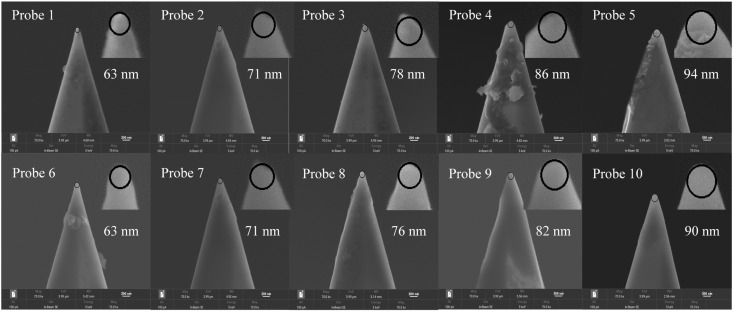
Wear tip morphology of Probes 1~10.

**Figure 5 sensors-23-04084-f005:**
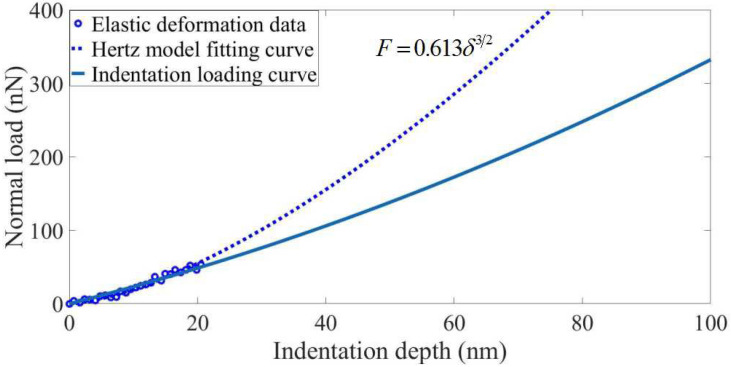
Hertz model fitting of indentation data for wear Probe 2.

**Figure 6 sensors-23-04084-f006:**
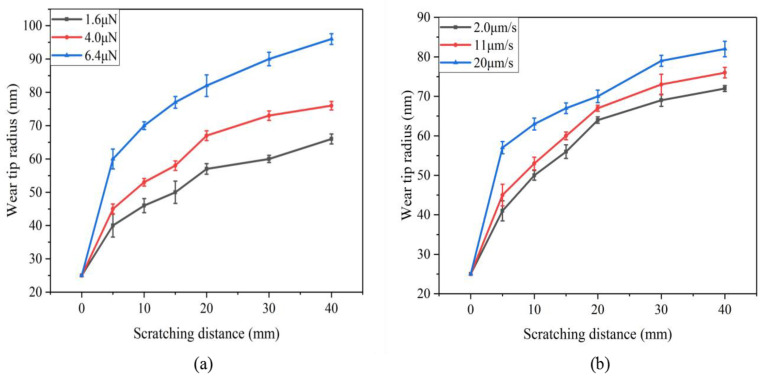
Relationship between scratching distance and wear tip radius: (**a**) different normal load levels; (**b**) different scratching speed levels.

**Figure 7 sensors-23-04084-f007:**
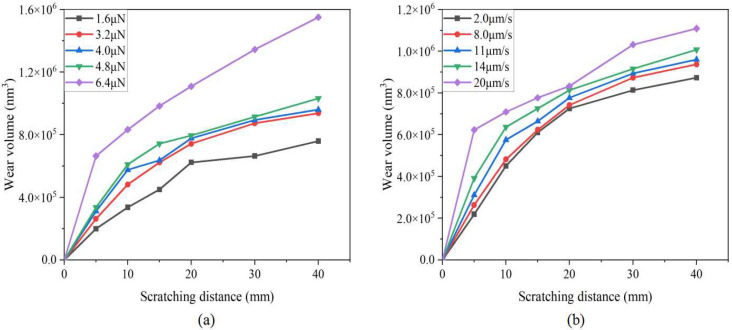
Relationship between scratching distance and probe wear volume: (**a**) different normal load levels; (**b**) different scratching speed levels.

**Figure 8 sensors-23-04084-f008:**
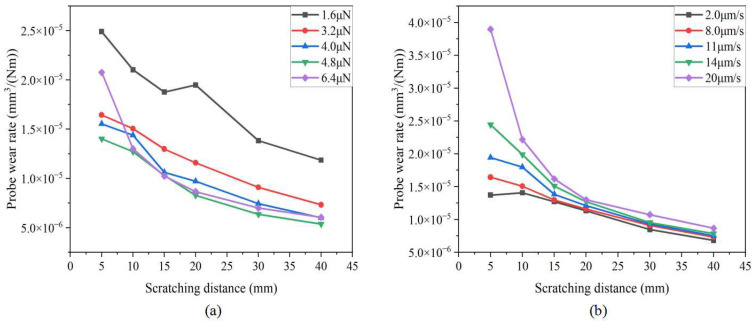
Relationship between scratching distance and probe wear rate: (**a**) different normal load levels; (**b**) different scratching speed levels.

**Figure 9 sensors-23-04084-f009:**
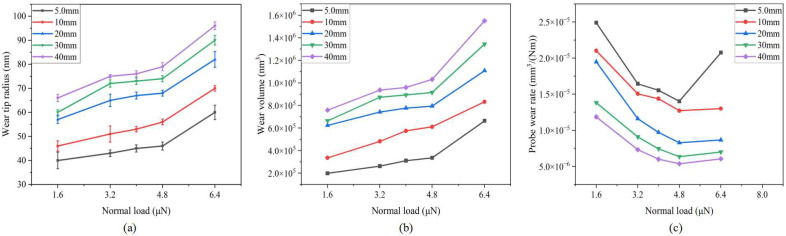
Relationship between normal load and evaluation index of probe wear state: (**a**) wear tip radius; (**b**) wear volume; (**c**) probe wear rate.

**Figure 10 sensors-23-04084-f010:**
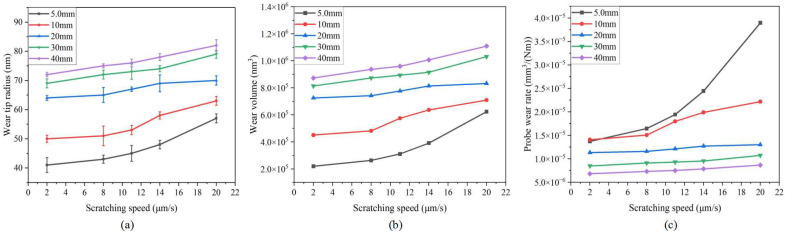
Relationship between scratching speed and evaluation index of probe wear state: (**a**) wear tip radius; (**b**) wear volume; (**c**) probe wear rate.

**Figure 11 sensors-23-04084-f011:**
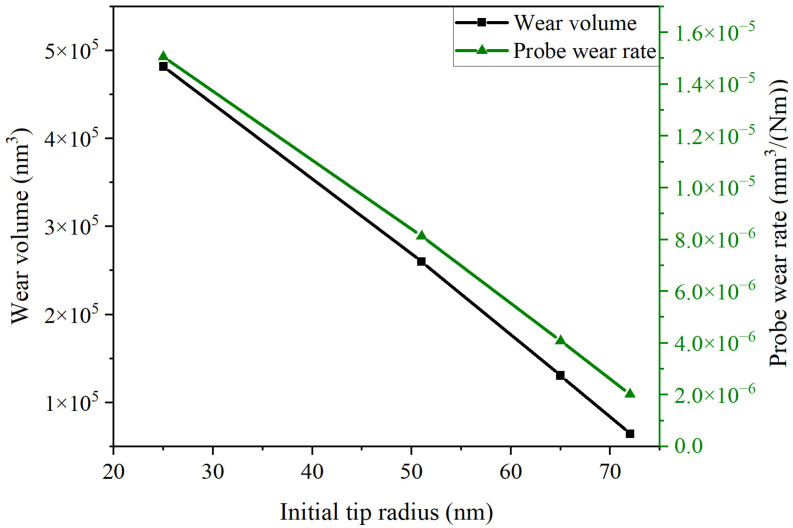
Relationship between the initial tip radius and the probe wear volume and probe wear rate.

**Figure 12 sensors-23-04084-f012:**
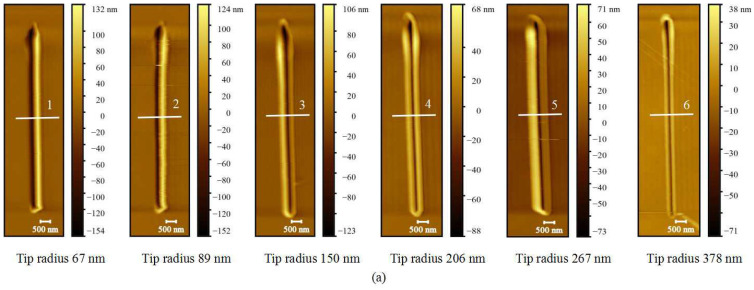
Grooves scratched by six wear probes with different tip radii: (**a**) grooves’ morphology; (**b**) grooves’ cross-section information; (**c**) variation in wear tip radius and depth of grooves with scratching distance.

**Figure 13 sensors-23-04084-f013:**
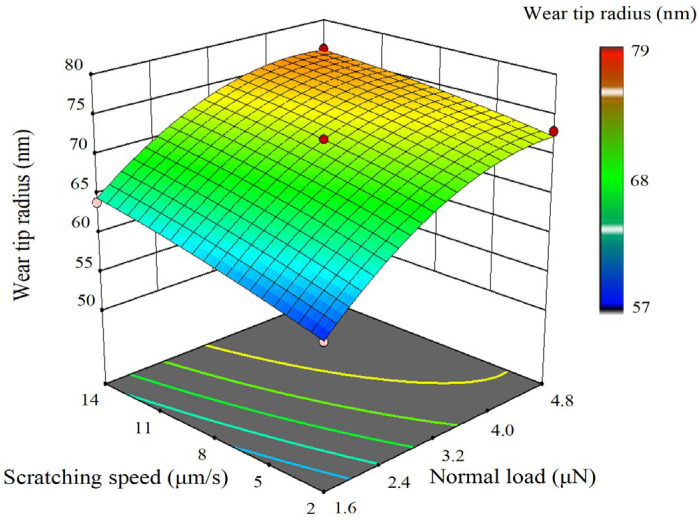
Three-dimensional response surface of wear tip radius.

**Figure 14 sensors-23-04084-f014:**
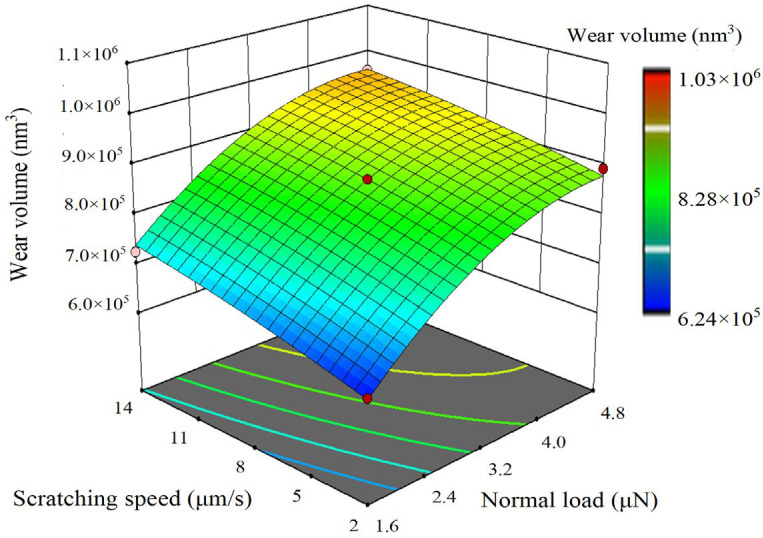
Three-dimensional response surface of wear volume.

**Figure 15 sensors-23-04084-f015:**
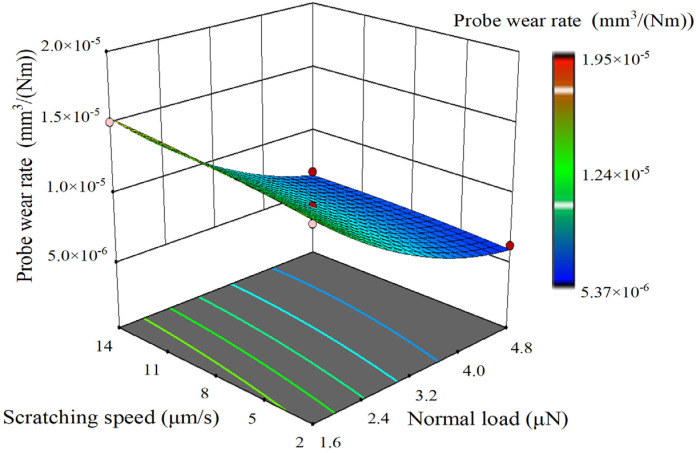
Three-dimensional response surface of probe wear rate.

**Table 1 sensors-23-04084-t001:** Nanomachining parameter setting in the experiment.

Probe Number	Normal Load (μN)	Scratching Speed (μm/s)	Scratching Distance (mm)	Experimental Purpose
Probe 1	1.6	8	40	Normal load effect
Probe 2	3.2	8	40	
Probe 3	4.0	8	40	
Probe 4	4.8	8	40	
Probe 5	6.4	8	40	
Probe 6	3.2	2	40	Scratching speed effect
Probe 7	3.2	8	40	
Probe 8	3.2	11	40	
Probe 9	3.2	14	40	
Probe 10	3.2	20	40	

**Table 2 sensors-23-04084-t002:** Calculation results and error analysis of wear tip radius of Probes 1~10.

Probe Number	Tip radius Measurement (nm)	Fitting Coefficient	Calculated Value of Tip Radius (nm)	Error Value
Probe 1	63	0.57	66	4.8%
Probe 2	71	0.61	75	5.6%
Probe 3	78	0.62	76	2.6%
Probe 4	86	0.63	79	8.1%
Probe 5	94	0.69	96	2.1%
Probe 6	63	0.60	72	14.3%
Probe 7	71	0.61	75	5.6%
Probe 8	76	0.62	76	0.0%
Probe 9	82	0.62	78	4.9%
Probe 10	90	0.64	82	8.9%

**Table 3 sensors-23-04084-t003:** Response surface experiment scheme and result.

Experiment Number	Normal Load (μN)	Scratching Speed (μm/s)	Scratching Distance (mm)	Wear Tip Radius (nm)	Wear Volume (nm^3^)	Probe Wear Rate (mm^3^/(Nm))
1	3.2	8	30	72	8.73 × 10^5^	9.09 × 10^−6^
2	4.8	8	20	68	7.95 × 10^5^	8.28 × 10^−6^
3	4.8	8	40	79	1.03 × 10^6^	5.37 × 10^−6^
4	4.8	14	30	76	9.60 × 10^5^	6.67 × 10^−6^
5	1.6	2	30	58	6.37 × 10^5^	1.33 × 10^−5^
6	1.6	14	30	64	7.26 × 10^5^	1.51 × 10^−5^
7	4.8	2	30	73	8.94 × 10^5^	6.21 × 10^−6^
8	3.2	14	20	69	8.14 × 10^5^	1.27 × 10^−5^
9	1.6	8	40	66	7.59 × 10^5^	1.19 × 10^−5^
10	3.2	14	40	78	1.01 × 10^6^	7.87 × 10^−6^
11	1.6	8	20	57	6.24 × 10^5^	1.95 × 10^−5^
12	3.2	2	20	64	7.26 × 10^5^	1.13 × 10^−5^
13	3.2	2	40	72	8.73 × 10^5^	6.82 × 10^−6^

**Table 4 sensors-23-04084-t004:** Analysis of variance for wear factors.

Response	Wear Factor	Sum of Squares	Mean Square Error	*F* Value	*p* Value	Significance
Wear tip radius	Functional model	586.17	65.13	71.05	0.0025	significant
	A—Normal load	325.12	325.12	354.68	0.0003	
	B—Scratching speed	50.00	50.00	54.55	0.0051	
	C—Scratching distance	171.13	171.13	186.68	0.0008	
	A^2^	32.14	32.14	35.06	0.0096	
Wear volume	Functional model	2.04 × 10^11^	2.26 × 10^10^	69.94	0.0025	significant
	A—Normal load	1.09 × 10^11^	1.09 × 10^11^	338.07	0.0004	
	B—Scratching speed	1.78 × 10^10^	1.78 × 10^10^	54.95	0.0051	
	C—Scratching distance	6.37 × 10^10^	6.37 × 10^10^	196.67	0.0008	
	A^2^	8.39 × 10^9^	8.39 × 10^9^	25.94	0.0146	
Probe wear rate	Functional model	2.03 × 10^−10^	2.25 × 10^−11^	87.67	0.0018	significant
	A—Normal load	1.38 × 10^−10^	1.38 × 10^−10^	536.3	0.0002	
	B—Scratching speed	2.80 × 10^−12^	2.80 × 10^−12^	10.89	0.0457	
	C—Scratching distance	4.95 × 10^−11^	4.95 × 10^−11^	192.63	0.0008	
	AC	5.56 × 10^−12^	5.56 × 10^−12^	21.64	0.0187	
	A^2^	4.42 × 10^−12^	4.42 × 10^−12^	17.23	0.0254	

**Table 5 sensors-23-04084-t005:** Coefficients of the theoretical formula.

Coefficient	Theoretical Model of Wear Tip Radius	Theoretical Model of Wear Volume	Theoretical Model of Probe Wear Rate
*z* _0_	21.4	1.463 × 10^5^	4.200 × 10^−5^
*a*	13	1.817 × 10^5^	−7.991 × 10^−6^
*b*	0.76	7.633 × 10^3^	3.312 × 10^−7^
*c*	0.78	8.210 × 10^3^	−9.325 × 10^−7^
*d*	−0.08	−5.885 × 10^2^	−3.620 × 10^−8^
*e*	0.03	1.588 × 10^3^	7.366 × 10^−8^
*f*	0.004	1.925 × 10^2^	−1.338 × 10^−9^
*g*	−1.46	−2.367 × 10^4^	5.434 × 10^−7^
*k*	−0.014	−2.292 × 10^2^	−4.795 × 10^−9^
*l*	−0.0075	−9.850 × 10^1^	7.649 × 10^−9^

## Data Availability

The dataset supporting the conclusions of this article is not available due to privacy and ethical reasons.

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
