# Peer review of "Experimental Investigation of Tip Wear of AFM Monocrystalline Silicon Probes"

_sensors, 2023, doi:10.3390/s23084084_

Round 1

Reviewer 1 Report

This study modelled the effects of load and speed on AFM probe wear. The reviewer feels that the experimental results of this study are valid, but are not of a high enough standard to be published in Sensors in terms of novelty and usefulness. I would like the authors to comment and revise the paper on the following points.

(1) The amount of experimental data on probe wear is too small. There are five levels of load experiments and five levels of scratching speed experiments, yet only one experiment was conducted at each level. In my opinion, the problem is that the data obtained is not verified for reproducibility by statistical processing. In addition, for the Error values shown in Table 2, statistical data should be given by multiple measurements under the same conditions.

(2) The results obtained show that increasing the load and scratching speed causes wear, but these results are not so new findings. The nanoindentation Hertz model  characterization method itself is also not a new method and it is unclear where the novelty lies. It is also unclear where the novelty lies, as it is not a new method and the inventiveness (not the differences) compared to the previous studies listed in references 11 and 21-26 is unclear.

(3) It is unclear in which situations the results of this research will be useful and the scope of its utility. For example, what kind of experimental data is needed to predict the degree of probe wear? What are the effects of different equipment conditions and probe materials? If only the trends observed under very limited equipment and operating conditions are described, the results are not particularly new and the usefulness of this research is considered to be very limited. It is interesting to note that a model equation that can reasonably predict the wear condition of the probe has been established, but the experimental data should make clear what this is useful for.

(4) Redundancy due to repeated use of data is found in the figures and tables. They should be more compact. For example, Figures 3 and 4 are duplicated. In addition, Figures 3-7 are merely different forms of representation of the same data.

(5) Basic information sufficient for reproduction, such as the equipment type used in the experiment and the source of the probes, is missing.

Reviewer 2 Report

The paper “Experimental Investigation of Tip Wear of AFM Monocrystalline Silicon Probe” by Song Huang, Yanling Tian, and Tao Wang describes a topic of general interest in nano technology. AFM technique was pioneer for catalysing nano technology for broad applications. Especially, it can be used under ambient conditions.    
The results were interpreted according to a model which was proposed by the famous Heinrich Hertz. Perhaps a sketch of the tip should be given. Fig 1 evaluates nearly no changes for non experts. Fig 2 shows no significant alteration, only debris can be seen. Instrumental settings cannot be seen!
Fig. 3 gives wear tip radius as function of scratching distance. However, no pictures of the tips are shown which will lead to these data. This is also true for the other figures. Thus, pictures should be given which make it possible to evaluate the tip radius. This is a central point of whole the paper.

Round 2

Reviewer 1 Report

This paper has been considerably improved by the revisions. Also, as many, if not all, of the questions have been answered, I would support the acceptance of this paper for Sensors.

Reviewer 2 Report

Now, the paper can be published as it is!
